# The Regulatory Effect of Receptor-Interacting Protein Kinase 3 on CaMKIIδ in TAC-Induced Myocardial Hypertrophy

**DOI:** 10.3390/ijms241914529

**Published:** 2023-09-26

**Authors:** Jianan Qian, Jingjing Zhang, Ji Cao, Xue Wang, Wei Zhang, Xiangfan Chen

**Affiliations:** 1School of Pharmacy, Nantong University, Nantong 226001, China; qjn29103@163.com (J.Q.); zhangjingjing@stmail.ntu.edu.cn (J.Z.); 2019310033@stmail.ntu.edu.cn (J.C.); w18862936086@163.com (X.W.); 2School of Medicine, Nantong University, Nantong 226001, China

**Keywords:** RIPK3, CaMKIIδ, necroptosis, myocardial hypertrophy, alternative splicing

## Abstract

Necroptosis is a newly discovered mechanism of cell death, and its key regulatory role is attributed to the interaction of receptor-interacting protein kinases (RIPKs) RIPK1 and RIPK3. Ca^2+^/calmodulin-dependent protein kinase (CaMKII) is a newly discovered RIPK3 substrate, and its alternative splicing plays a fundamental role in cardiovascular diseases. In the present study, we aimed to explore the role and mechanism of necroptosis and alternative splicing of CaMKIIδ in myocardial hypertrophy. Transverse aortic constriction (TAC) was performed on wild-type and knockout mice to establish the model of myocardial hypertrophy. After 3 weeks, echocardiography, cardiac index, cross-sectional area of myocardial cells, hypertrophic gene expression, myocardial damage, and fibers were assessed. Moreover, we detected the levels of inflammatory factors (IL-6 and TNF-α) and examined the expressions of necroptosis-related proteins RIPK3, RIPK1, and phosphorylated MLKL. Meanwhile, we tested the expression levels of splicing factors ASF/SF2 and SC-35 in an attempt to explore CaMKII δ. The relationship between variable splicing disorder and the expression levels of splicing factors ASF/SF2 and SC-35. Further, we also investigated CaMKII activation, oxidative stress, and mitochondrial ultrastructure. In addition, wild-type mice were administered with a recombinant adeno-associated virus (AAV) carrying RIPK3, followed by TAC surgery to construct a model of myocardial hypertrophy, and the above-mentioned indicators were tested after 3 weeks. The results showed that RIPK3 deficiency could alleviate cardiac dysfunction, myocardial injury, aggravation of necrosis, and CaMKII activation induced by TAC surgery in mice with myocardial hypertrophy. Tail vein injection of AAV could reverse cardiac dysfunction, myocardial damage, aggravation of necrosis, and CaMKII activation in mice with myocardial hypertrophy. These results proved that RIPK3 could be used as a molecular intervention target for the prevention and treatment of myocardial hypertrophy.

## 1. Introduction

Myocardial hypertrophy is a compensatory mechanism of the heart’s stress response to pressure and volume overload. The early stage of myocardial hypertrophy induced by pressure overload is characterized by an increased volume of myocardial cells, enhanced protein synthesis, and elevated myocardial contractility. After long-term pathological stress, myocardial hypertrophy is gradually increased, and then myocardial fibrosis, sparse capillaries, oxidative stress, inflammation, cell death, and cardiac dysfunction occur [1,2]. Continuous excessive workload will lead to the breakdown of the compensatory mechanism, and the final fate of the development of myocardial hypertrophy is heart failure, which is an independent risk factor for cardiovascular disease [3,4]. However, up to now, the molecular mechanism underlying myocardial hypertrophy is still poorly understood. Therefore, it is highly necessary to identify more effective regulatory targets for the prevention and treatment of myocardial hypertrophy.

In recent years, researchers have discovered a new form of cell death that does not depend on the activation of the caspase family, namely necroptosis, which has both the characteristics of cell apoptosis and cell necrosis [5,6,7]. Necroptosis is mainly manifested as cell enlargement, organelle swelling, organelle dysfunction, plasma membrane rupture, the release of intracellular components, and no significant changes in nuclear chromatin [6,8]. Several studies have shown that receptor-interacting protein kinase 3 (RIPK3) is a key regulator in the necroptosis signaling pathway, and necroptosis is heavily dependent on the necrotic bodies formed by the interaction between RIPK1 and its homolog of RIPK3 [9]. RIPK3 directly acts upstream of the pseudokinase mixed-lineage kinase domain-like protein (MLKL) in the signal transduction cascade, leading to necrosis [10], prompting its phosphorylation and activation to form MLKL oligomers. The N-terminal of MLKL can destroy lipid vesicles, directly penetrate the plasma membrane, and ultimately promote cell death [7]. Therefore, MLKL, as the effector of RIPK3, is the key executive molecule of the bad apoptosis signaling pathway [11]. Current studies believe that RIPK-dependent necroptosis is related to the occurrence of several cardiovascular diseases, such as myocardial ischemia-reperfusion injury [12], atherosclerosis [13], and diabetic cardiomyopathy [6]. However, the role of RIPK3-mediated necroptosis in myocardial hypertrophy caused by pressure overload remains largely unexplored.

Meanwhile, some studies have found that in the diseased myocardial tissue, the occurrence of necroptosis is not only dependent on the RIPK1/RIPK3/MLKL cascade reaction but also regulated by the RIPK3/CaMKII signaling pathway [14,15]. Ca^2+^-calmodulin-dependent protein kinase II (CaMKII) is a serine/threonine kinase with multiple functions, such as regulation of Ca^2+^ processing, cell-to-cell coupling, cell death, inflammation, and mitochondrial function [16,17,18]. The latest research has found that CaMKII is a new substrate of RIPK3 [19,20], and RIPK3 activates CaMKII protein by directly phosphorylating CaMKII protein at the T287 site (autophosphorylation site) or indirectly oxidizing CaMKII by increasing reactive oxidative species (ROS) production. Meanwhile, studies have found that CaMKII activation can lead to the opening of mitochondrial permeability transition pores (mPTP) and cardiomyocyte death [14,20,21]. Zhang et al. [14] have demonstrated that RIPK3-mediated CaMKII activation is involved in ischemia and oxidative stress-induced myocardial necroptosis cell death. More and more evidence also shows that the continuous activation of CaMKII plays a central role in the pathogenesis of a variety of heart diseases. Therefore, it is highly necessary to clarify the role and mechanism of CaMKII in pressure overload-induced cardiac hypertrophy for the prevention and treatment of cardiac hypertrophy.

As a key transduction factor of calcium signal in the heart, CaMKII includes four subtypes: α, β, γ, and δ, and CaMKIIδ is the main subtype in the heart [21,22]. The precursor exons 14, 15, and 16 of CaMKIIδ mRNA are regulated using various splicing factors of the SR protein family, such as ASF/SF2 and SC-35, and then three CaMKIIδ variants are generated: CaMKIIδ A, CaMKIIδ B, and CaMKIIδ C [21]. The functions of the three CaMKIIδ variants are not the same. Relevant studies have shown that CaMKIIδ A mainly mediates myocardial excitation-contraction coupling [23], CaMKIIδ B is mainly distributed in the nucleus [24], and CaMKIIδ C modifies a variety of Ca^2+^ mainly through phosphorylation. Regulatory proteins play an important role in the coupling of myocardial excitation and contraction [25]. The alternative splicing of CaMKIIδ is strictly regulated. Once the alternative splicing of CaMKIIδ is out of balance, it will cause cardiomyocyte dysfunction and eventually lead to heart disease [26]. Recent studies have shown that inhibiting RIPK3 can alleviate the disorder of CaMKIIδ alternative splicing in cardiomyocytes stimulated by high glucose [6]. However, whether CaMKIIδ alternative splicing also plays a similar role in myocardial hypertrophy is still unknown.

Therefore, in the present study, we aimed to explore the role and mechanism of RIPK3-mediated necroptosis in pressure overload-induced cardiac hypertrophy. Through the tail vein injection of RIPK3^−/−^ mice and wild-type (WT) mice with RIPK3 shRNA recombinant adeno-associated virus (AAV), transverse aortic constriction (TAC) surgery was performed to explore the effects of RIPK3 deficiency and down-regulation on myocardial hypertrophy. We also tried to prove that silencing and down-regulation of RIPK3 could regulate CaMKIIδ alternative splicing and CaMKII activity to delay the pathogenesis of cardiac hypertrophy.

## 2. Results

### 2.1. Cardiac Hypertrophy in Mice with Cardiac Dysfunction, Necroptosis, and CaMKIIδ Alternative Splicing Disorder

The cross-sectional area of mouse cardiomyocytes was observed using WGA staining (Figure 1A). The results showed that TAC surgery significantly increased the cross-sectional area of mouse cardiomyocytes in WT mice (Figure 1A,B). Meanwhile, after TAC surgery, the expressions of myocardial hypertrophy genes ANP and BNP in WT mice were significantly increased (Figure 1C–F), suggesting that a mouse model of myocardial hypertrophy was successfully established using TAC surgery. Several studies have shown that RIPK3 is involved in cell necroptosis. Our study found that the expression of RIPK3 in the myocardial tissue of mice with myocardial hypertrophy was significantly increased (Figure 2A). Overexpression of RIPK3 simultaneously up-regulated the expression of RIPK1 and the phosphorylation of MLKL in the myocardial tissues of mice with cardiac hypertrophy (Figure 2B,C). In addition, apoptosis is a form of cell necrosis. The expression of cleaved caspase-3 in the hypertrophic group was significantly higher compared with the sham operation group (Figure 2D). CaMKIIδ alternative splicing disorder easily promotes cardiomyocyte dysfunction and ultimately leads to heart disease. Since there is no specific antibody for CaMKIIδ variation, qRT-PCR was used to detect the expressions of CaMKIIδ A, CaMKIIδ B, and CaMKIIδ C at the mRNA level. CaMKIIδ A and CaMKIIδ B in diabetic mice were significantly reduced, and the expression of CaMKIIδ C was significantly increased, indicating that the alternative splicing of CaMKIIδ in mice with myocardial hypertrophy was disordered. In addition, the oxidation and phosphorylation of CaMKII were also increased in the TAC group (Figure 2E,F). Our study showed cardiac dysfunction in mice with myocardial hypertrophy, such as myocardial tissue necroptosis and CaMKIIδ alternative splicing disorder (Figure 2G).

### 2.2. RIPK3 Deficiency Can Alleviate Cardiac Hypertrophy, Myocardial Injury, Myocardial Fibrosis, and Inflammation

We used RIPK3^−/−^ mice to verify the role of RIPK3 in the development of myocardial hypertrophy. The cross-sectional area of mouse cardiomyocytes was observed using WGA staining, proving that both WT and RIPK3^−/−^ mice developed cardiac hypertrophy after TAC surgery (Figure 3A,B). Moreover, the depletion of the RIPK3 gene did not affect the expressions of the hypertrophic genes in mice with myocardial hypertrophy (Figure 3C–F). The measurement and statistics of mouse heart index IVS, LVPW, HW, HMI, LVMI, and LVMI/TL also proved that depletion of RIPK3 did not affect the degree of myocardial hypertrophy induced by TAC surgery in mice (Figure 4C,G–J). However, in mice with myocardial hypertrophy, the lack of RIPK3 significantly increased EF and FS, indicating that the cardiac function of the mice was improved (Figure 4A,B). H&E staining showed that depletion of RIPK3 alleviated the distortion and arrangement disorder of cardiomyocytes in mice with cardiac hypertrophy (Figure 4D–F). We used Sirius Red and Masson staining to evaluate the degree of myocardial fibrosis in mice. The test results showed that depletion of RIPK3 significantly improved the degree of collagen deposition and myocardial fibrosis (Figure 5A,B). The levels of serum LDH and CK were detected to evaluate the degree of myocardial injury. After TAC surgery, the levels of serum LDH and CK in RIPK3^−/−^ mice were significantly lower compared with WT mice. Depletion of RIPK3 improved myocardial injury in mice with cardiac hypertrophy (Figure 5C,D). The levels of serum IL-6 and TNF-α were detected using ELISA, showing that the level of inflammation was significantly reduced in the myocardial tissue in the RIPK3^−/−^ group, indicating that the myocardial injury was improved to a certain extent (Figure 5E,F). In summary, although depletion of RIPK3 did not change the degree of myocardial hypertrophy, it could significantly improve the central dysfunction of myocardial hypertrophy, such as myocardial injury, myocardial fibrosis, and inflammation.

### 2.3. Loss of RIPK3 can Ameliorate Myocardial Necroptosis in Mice with Cardiac Hypertrophy, Regulate CaMKIIδ Splicing Disorder, Improve Oxidative Stress, and Ameliorate Myocardial Mitochondrial Ultrastructure

In addition to RIPK3, studies have shown that RIPK1 is also involved in the process of necrosis, and the phosphorylation of MLKL is an essential effector molecule for necroptosis. Western blotting analysis showed that the expressions of RIPK1 and phosphorylated MLKL in the myocardium of RIPK3^−/−^ mice with myocardial hypertrophy were significantly lower compared with WT mice (Figure 6A–C). Moreover, apoptosis was significantly improved in RIPK3^−/−^ mice with myocardial hypertrophy (Figure 6D,G). A recent study has found that CaMKII is one of the substrates of RIPK3 in cardiac ischemic diseases. Our results showed that depletion of RIPK3 could attenuate the oxidation and phosphorylation of CaMKII in the myocardium of mice with myocardial hypertrophy (Figure 6E,F). Meanwhile, the expressions of CaMKIIδ variants A, B, and C were detected using PCR, and depletion of RIPK3 improved the CaMKIIδ mutation to a certain extent (Figure 6H). As we know, studies have found that ASF/SF2 and SC-35 are classic SR proteins that play an important role in alternative splicing. Therefore, we used Western blotting to detect the expressions of ASF/SF2 and SC-35 mice with myocardial hypertrophy to explore whether they play a decisive role in alternative splicing of CaMKIIδ. Western blotting analysis showed that the expressions of ASF and SC-35 in the myocardium of RIPK3^−/−^ mice with myocardial hypertrophy were significantly lower compared with WT mice. (Figure 6 I,J) In addition, oxidative stress plays an important role in the occurrence and development of heart disease. ROS accumulation is the main factor in myocardial necrosis, which may reduce the survival of myocardial cells and aggravate myocardial damage. We used DHE staining to evaluate the level of ROS in tissues and found that at 3 weeks after the TAC surgery, the red fluorescence intensity of myocardial tissue in WT mice was increased, while the red fluorescence intensity of myocardial tissue in RIPK3^−/−^ mice was weaker compared with WT mice, indicating that depletion of RIPK3 reduced the accumulation of ROS in myocardial tissue (Figure 7A,B). We selected MDA, T-AOC, and T-SOD kits to evaluate the AOC of mouse myocardial tissue, and the results showed that the myocardial tissue’s ability to scavenge oxygen free radicals was enhanced in RIPK3^−/−^ mice, and the accumulation of ROS was decreased (Figure 7C–E). The ultrastructure of myocardial mitochondria in the left ventricle of mice with myocardial hypertrophy was observed using a transmission electron microscope. The mitochondria were irregularly swollen, and cristae fractured. However, in RIPK3^−/−^ mice, the mitochondrial abnormalities of DCM were improved (Figure 7F).

### 2.4. AAV-RIPK3 shRNA Interferes with the Expression of RIPK3, Reverses the Dysfunction of Myocardial Hypertrophy, Myocardial Fibrosis, and Inflammation, and Reduces Myocardial Damage

To further evaluate the relative role of RIPK3 in myocardial hypertrophy, we injected recombinant AAV-carrying RIPK3 shRNA directly into the tail vein of mice to interfere with the expression of RIPK3. After TAC surgery, the cross-sectional area of myocardial cells in each group was significantly increased, and the expressions of hypertrophic genes were increased (Figure 8D–I). Echocardiography showed that after interfering with the expression of RIPK3, IVS, and LVPW were not changed significantly, while the values of EF and FS were significantly decreased, and the cardiac contractile function was improved (Figure 8A–C). Meanwhile, the cardiomyocytes were twisted, and the disordered arrangement was reversed. Through Sirius red and Masson staining, we found that after interfering with RIPK3, myocardial collagen deposition in mice with myocardial hypertrophy was reduced, and myocardial fibrosis was improved (Figure 9A). At the same time, myocardial tissue damage and inflammation were reversed (Figure 9B–E). Taken together, these findings showed that interference with RIPK3 could indeed prevent myocardial damage in mice with myocardial hypertrophy. 

### 2.5. AAV-RIPK3 shRNA Interference Relieves Myocardial Necroptosis in Mice with Myocardial Hypertrophy, Regulates CaMKIIδ Splicing Disorder, Improves Oxidative Stress, and Ameliorates Myocardial Mitochondrial Ultrastructure

After injecting recombinant AAV into the tail vein, the expression of RIPK3 was decreased. In mice with cardiac hypertrophy, the expression of RIPK3 downstream protein RIPK1 and the phosphorylation of MLKL were inhibited, and meanwhile, apoptosis was improved (Figure 10A–D,G). In addition, AAV-RIPK3 shRNA directly inhibited the levels of oxidation and phosphorylation of its downstream necroptosis-related effector CaMKII (Figure 10E,F). Moreover, the expression of the CaMKIIδ variant was corrected (Figure 10H). Additionally, we found the abnormal expression of ASF and SC-35 were suppressed. (Figure 10I,J) Further, DHE staining (Figure 11A–E) and transmission electron microscopy (Figure 11F), respectively, showed that after the interference of RIPK3, the oxidative stress level and the abnormalities of mitochondrial ultrastructure in mice with myocardial hypertrophy were corrected. 

## 3. Discussion

Myocardial hypertrophy includes physiological hypertrophy and pathological hypertrophy. Persistent pathological myocardial hypertrophy can lead to congestive heart failure, arrhythmia, and sudden death [27,28]. Myocardial hypertrophy is often accompanied by cardiac dysfunction and myocardial damage. The characteristics of myocardial hypertrophy are myocardial fibrosis, oxidative stress damage, inflammatory response, and cellular dysfunction [29,30]. In the present study, we successfully established a mouse model of pressure-overload-type cardiac hypertrophy using TAC surgery. We found that the values of EF and FS in the heart of mice with myocardial hypertrophy were decreased, the activities of serum LDH and CK were increased, the deposition of myocardial collagen was increased, the levels of MDA, T-AOC, and T-SOD in myocardial tissue were increased, and the levels of serum IL-6 and TNF-α were elevated. As a newly discovered type of cell death, many studies have proved that necroptosis plays an important role in the occurrence and development of a variety of cardiovascular diseases [31]. Our experiments showed that cardiomyocyte necroptosis in mice with myocardial hypertrophy was aggravated. Given that RIPK3 was a key signaling molecule in the pathways related to necroptosis, we studied the differences in cardiac hypertrophy between WT mice and RIPK3^−/−^ mice. Moreover, we tried to prove that down-regulation and depletion of RIPK3 could alleviate myocardial necroptosis to a certain extent and improve heart function.

Necroptosis, cell death caused by uncontrolled swelling (tumor) and rapid plasma membrane rupture, has harmful pathogenic effects [32]. RIPK3 is the convergence point of a variety of signaling pathways, including necrosis, inflammation, and oxidative stress. The interaction of the motif (RHIM) domain enters the complex and promotes the formation of necrotic bodies [33]. The formation of necrotic bodies results in the activation (phosphorylation) of MLKL and its subsequent plasma membrane translocation. Once MLKL is located in the plasma membrane, permeable pores may be formed, leading to the destruction of the integrity of the plasma membrane, cell death, and necrosis [34,35]. In addition, TNF-α has been shown to play a key role in the initiation and induction of necrosis. TNF-α can activate cell suicide programs, such as apoptosis and necrosis, and cause the accumulation and phosphorylation of RIPK1 and RIPK3, leading to phosphorylation of MLKL and increased formation of necrotic bodies and necrosis [36]. On the other hand, the Deepa SS’s [37] team has found that the increased necroptosis is accompanied by an increase in a variety of inflammatory cytokines, including the pro-inflammatory cytokines IL-6 (3.9 times) and TNF-α (4.7 times). Our results showed that the expression of RIPK1 and the phosphorylation of MLKL were significantly up-regulated in the myocardium of WT mice with myocardial hypertrophy, and the levels of serum IL-6 and TNF-α were increased. In the TAC group, the expression of RIPK1 in RIPK3^−/−^ mice was decreased, the phosphorylation of MLKL was decreased, and the levels of serum IL-6 and TNF-α were decreased. After injection of the AAV-carrying RIPK3 into the tail vein, the effect was the same as that of RIPK3^−/−^ mice, indicating that the specific pathway of RIPK3-induced myocardial necroptosis was closely related to RIPK1 and MLKL.

CaMKII is a multifunctional serine/threonine-protein kinase [21], a polymer compound composed of 12 monomers [38,39]. It can be phosphorylated and regulated. As the key protein of excitation-contraction coupling (ECC) and excitation-transcription coupling (ETC) [40], CaMKII is a pleiotropic signal that regulates cardiomyocyte Ca^2+^ circulation, contraction, inflammation, metabolism, gene expression, and cell survival [20]. Recent studies have shown that CaMKII is a new substrate of RIPK3 [14,19]. RIPK3 constitutes an important upstream kinase of CaMKII in the mechanism. There are at least two pathways involved in RIPK3-mediated CaMKII activation, including direct phosphorylation and indirect ROS-mediated oxidation, which trigger the opening of mPTPs and myocardial necrosis [14,20,21]. Studies have reported the continuous activation of CaMKII is the central intracellular trigger of various heart diseases [40]. There is evidence that sustained CaMKII activation is involved in a large number of major heart conditions, such as heart failure [41], arrhythmia [42], and sudden cardiac death [43]. During excitation-contraction coupling (ECC), CaMKII phosphorylates several Ca-handling proteins, including ryanodine receptors (RyR), phospholamban, and L-type Ca channels. CaMKII expression and activity have been shown to correlate positively with impaired ejection fraction in the myocardium of patients with heart failure, and CaMKII has been proposed to be a possible compensatory mechanism to keep hearts from complete failure. However, in addition to these acute effects on ECC, CaMKII was shown to be involved in hypertrophic signaling, termed excitation-transcription coupling (ETC). Thus, animal models have shown that overexpression of nuclear isoform CaMKIIdeltaB can induce myocyte hypertrophy [22]. Our study found that in WT mice with myocardial hypertrophy, the phosphorylation and oxidation of CaMKII were significantly enhanced, while these conditions were significantly reversed in RIPK3^−/−^ mice. After injection of the AAV-carrying RIPK3 into the tail vein, the effect was the same as that of RIPK3^−/−^ mice. In summary, the CaMKII signaling pathway might be the downstream or substrate of RIPK3 in myocardial hypertrophy. CaMKII overexpression can lead to myocyte hypertrophy and heart failure and CaMKII-dependent phosphorylation and oxidation of CaMKII.

CaMKII has four isomers (α, β, γ, and δ) in different types of tissues with different expression rates. α and β are mainly distributed in neurons, and δ and part of γ are mainly distributed in cardiomyocytes. At present, the research on CaMKIIδ is the most and most comprehensive [38]. CaMKIIδ expresses three variants of CaMKIIδ A, CaMKIIδ B, and CaMKIIδ C after alternative splicing of its exons 14, 15, or 16 using splicing factors. Different subtypes of CaMKIIδ have different functions. CaMKIIδ A mediates the coupling of myocardial excitation and contraction. CaMKIIδ B may even induce pathological cardiac remodeling using phosphorylation of histone deacetylase. CaMKIIδ C participates in cardiomyocyte apoptosis [21]. The variable shear of CaMKIIδ is strictly regulated. Once a disorder occurs, the expressions of the three variants are unbalanced, leading to the dysfunction of myocardial cells and ultimately resulting in heart disease [6]. ASF/SF2 and SC-35 are key factors regulating alternative splicing of CaMKIIδ pre-mRNA [44]. In cardiomyocytes, down-regulation of ASF/SF2 significantly decreased the expressions of CaMKIIδ C and simultaneously increased the expressions of CaMKIIδ A and CaMKIIδ B at the mRNA level [45]. SC-35 is another member of the SR protein family, and its alternative splicing effect on CaMKIIδ is similar to that of ASF/SF2 [46]. After the downregulation of RIP3 expression in NRCMs, the phenotypes of myocardial hypertrophy were obviously alleviated. Research has shown that RIP3 interacts with mixed lineage kinase domain-like protein (MLKL) and promotes its cell membrane localization to increase the influx of calcium within cells, thereby mediating the development of myocardial hypertrophy [47]. Our experimental results showed a key role of the RIP3-MLKL signaling pathway in myocardial hypertrophy. The RIP3-MLKL signaling pathway also plays a crucial role in myocardial hypertrophy. Our research results indicate that compared with the control group, the expressions of CaMKIIδ A and CaMKIIδ B at the mRNA level in the myocardial tissue of the WT TAC group were decreased, and the expression of CaMKIIδ C at the mRNA level was increased. This situation was corrected in the RIPK3^−/−^ TAC group and the AAV intervention group. The mechanism may be that inhibition of RIPK3 can affect the expression of splicing factors ASF and SC-35, thereby regulating alternative splicing disorders. Therefore, we considered that inhibition of RIPK3 could correct myocardial CaMKIIδ alternative splicing disorder in mice with cardiac hypertrophy.

Increased oxidative stress is one of the main pathophysiological mechanisms of many cardiovascular diseases [48]. There is evidence that the production of ROS during myocardial hypertrophy leads to necrosis. In addition, necroptosis can also exacerbate the production of ROS, suggesting that necroptosis and ROS form a vicious circle with each other [49]. Mitochondria are the main source of ROS and energy production. Once the mitochondria are damaged, their oxidative phosphorylation capacity is decreased, which can lead to excessive ROS production and abnormal calcium homeostasis, thereby promoting the death of cardiomyocytes [50]. We found that TAC induced mitochondrial ultrastructural disorder in the myocardial hypertrophy group, with obvious swelling of mitochondria, disordered arrangement, shortened mitochondrial cristae, and decreased number of mitochondria. Meanwhile, its oxidative stress level was significantly enhanced. However, such situations in the RIPK3^−/−^ group and the AAV infection group were improved significantly.

## 4. Materials and Methods

### 4.1. Animals

WT C57BL/6 mice (male, 8 weeks old) were provided by the Experimental Animal Center of Nantong University (Nantong, China). RIPK3^−/−^ mice were donated by the Institute of Molecular Medicine, Peking University (Beijing, China). The animals were housed in cages in the Experimental Animal Center of Nantong University at 20 °C with a 12 h light/dark cycle, and the mice were fed a standard laboratory diet and tap water. All procedures complied with the recommendations of the “Guidelines for the Care and Use of Laboratory Animals” (approval number: NTU-20161225) issued by the National Institutes of Health and the Animal Care and Use Steering Committee of Nantong University.

### 4.2. Establishment of a Mouse Model of Myocardial Hypertrophy 

WT C57BL/6 mice and RIPK3^−/−^ mice (18–22 g) were randomly assigned to the sham operation group and operation group. The animals were intraperitoneally anesthetized with ketamine (100 mg·kg ^−1^) and xylazine (5 mg·kg ^−1^) and monitored using the reflex of pinching the toes with forceps. The thoracic cavity was opened at the second intercostal space at the upper-left edge of the sternum via a small incision, and the animal was ventilated with a respirator. After the aortic arch was exposed, TAC was performed between the left common carotid artery and brachiocephalic artery by ligating a 6–0 nylon suture using a 27-gauge needle. After the needle was quickly withdrawn, an incomplete contraction was formed. In the sham operation group, the mice underwent the same surgery but did not shrink. At the end of the operation, the chest was closed with 5–0 nylon sutures. The mice were closely monitored for any signs of postoperative pain or dyspnea. In case of any signs of pain, animals received buprenorphine every 6–12 h as needed (TIPR Pharmaceutical Co., Ltd., Tianjin, China; 0.1–2.5 mg·kg^−1^, subcutaneous injection). There were no postoperative deaths in our experiments.

### 4.3. Tail Vein Injection of AAV

To evaluate the role of the RIPK3 gene in myocardial hypertrophy, male C57BL/6 mice (8 weeks old) were divided into two groups, and AAV-vector and AAV-RIPK3 shRNAs were injected via the tail vein. The injection dose was 1 × 10^11^ vg per mouse once, and TAC was performed one week later to establish a mouse myocardial hypertrophy model. The operation method was the same as the above-mentioned.

### 4.4. Echocardiography

At 3 weeks after the operation, the mice were anesthetized and maintained with isoflurane (1–2%). The heart geometry was measured at a probe frequency of 30 MHz from the parasternal long-axis view. M-mode echocardiography was used to record a clear image of the left ventricular area. The thickness of the interventricular septum (IVS) and left ventricle posterior wall (LVPW) was determined. Subsequently, the ejection fraction (EF) and left ventricular fractional shortening (FS) were calculated based on the average of 10 cardiac cycles.

### 4.5. Heart Index Determination

The blood sample was collected from the eyeball, and the serum sample was obtained after static centrifugation. After the blood was collected, the mice were killed with an overdose of isoflurane (5%). The heart was immediately removed and washed with pre-chilled saline to remove any blood clots. All connective tissue and blood vessels attached to the heart were also removed. The heart was dried with filter paper, and then the heart weight (HW) was determined with an electronic balance. The left ventricular weight (LVW), including the weight of IVS, was measured after removing the atrium and right ventricle. Heart mass index (HMI) and left ventricular mass index (LVMI) were calculated as the ratios of HW to body weight (BW) and LVW to BW, respectively. Tibia length (TL) was measured from the edge of the tibial plateau to the medial malleolus of the right hind limb. The ratio of LVW to TL was calculated and used as an indicator of cardiac hypertrophy.

### 4.6. Hematoxylin-Eosin (H&E) Staining

The left ventricles were fixed overnight in 4% paraformaldehyde, embedded in paraffin, and then cut into 5-μm sections. The sections were subjected to H&E staining and dehydrated with ethanol. Subsequently, the sections were observed and photographed under an optical microscope.

### 4.7. Sirius Scarlet Staining

The left ventricles were fixed overnight in 4% paraformaldehyde, embedded in paraffin, and then cut into 5-μm sections. After staining with Weigert’s iron hematoxylin staining solution and then drip staining with Sirius scarlet staining solution, the sections were conventionally dehydrated to transparent, sealed with neutral gum, and photographed under a microscope.

### 4.8. Masson Staining

The left ventricles were fixed overnight in 4% paraformaldehyde, embedded in paraffin, and then cut into 5-μm sections. The paraffin-embedded sections were deparaffinized in water, chromated with potassium dichromate overnight, stained with iron hematoxylin, ponceau acid fuchsin, phosphomolybdic acid, and aniline blue in turn, then dehydrated, mounted, and examined under a microscope.

### 4.9. WGA Staining

The left ventricles were fixed overnight in 4% paraformaldehyde, embedded in paraffin, and then cut into 5-μm sections. The paraffin-embedded sections were deparaffinized in water. After antigen retrieval, the WGA working solution was added dropwise, followed by incubation for 30 min. Subsequently, the sections were stained for nuclear mounting and examined under a microscope.

### 4.10. TUNEL Staining

The left ventricles were fixed overnight in 4% paraformaldehyde, embedded in paraffin, and then cut into 5-μm sections. After staining with TUNEL (Beyotime, Shanghai, China) at 37 °C for 60 min, the sections were washed three times with PBS, then observed, and photographed under an optical microscope. The quantification was performed using Image J version 1.51j8 (National Institutes of Health, USA) software.

### 4.11. Measurement of Superoxide Formation

The production of superoxide in myocardial tissue was detected using dihydroethidium (DHE) staining under a fluorescence microscope. Briefly, myocardial tissue sections (5 μm) were prepared and then incubated (30 min, 37 °C) in Krebs-HEPES buffer (mM components: NaCl 99, KCl 4.7, MgSO_4_ 1.2, KH_2_ PO_4_ 1.0, CaCl_2_ 1.9, NaHCO_3_ 25, glucose 11.1, Na HEPES 20; pH 7.4) containing 2 μM DHE in a dark room. The slides were examined with a Nikon TE2000 inverted microscope (Nikon, Tokyo, Japan) at excitation and emission wavelengths of 480 and 610 nm, respectively. 

The thiobarbituric acid method (Beyotime) was used to detect the level of malondialdehyde (MDA) in the myocardium. The total antioxidant capacity (T-AOC) of the myocardium was measured using the T-AOC detection kit using the plasma iron reduction capacity method (Beyotime). The activities of total SOD, Cu-Zn/SOD, and Mn-SOD in the myocardium were determined using the WST-1 (2-(4-iodophenyl)-3-(4-nitrophenyl)-5-(2,4-disulfophenyl)-)-2 H-tetrazole) method (Beyotime).

### 4.12. Determination of Blood Biochemical Indicators

Before sacrifice, whole blood was collected from the orbital vein of each mouse and centrifuged at 3000× *g* for 20 min. The serum was kept at −80 °C for other analysis. Corresponding commercial kits were used to detect the LDH activity and the contents of CK, IL-6, and tumor necrosis factor-α (TNF-α) in mouse serum.

### 4.13. Western Blotting Analysis

The proteins extracted from the myocardial tissue were subjected to SDS-PAGE and transferred onto the PVDF membranes (Millipore, Billerica, MA, USA). After blocking with TBST buffer (Tris-HCl 10 mmol·L^−1^, NaCl 120 mmol·L^−1^, Tween-20 0.1%; pH 7.4) containing 5% skimmed milk (*v*/*v*) at room temperature for 2 h, the membranes were incubated with primary antibodies at 4 °C overnight, including anti-ANP (1:1000, Abcam, Cambridge, UK), anti-BNP (1:1000, Abcam, Cambridge, UK), anti-ox-CaMKII (1:1000, Millipore, Kenilworth, NJ, USA), anti-CaMKII (1:1000, Abcam, Cambridge, UK), anti-caspase-3, anti-cleaved caspase-3, anti-MLKL, anti-p-MLKL,anti-RIPK3 and anti-RIPK1 (1:1000, Cell Signaling Technology, Danvers, MA, USA); anti-p-CaMKII (1:1000, Thermo Fisher Scientific, Rockford, IL, USA), anti-SRSF1(ASF/SF2), anti-SRSF2(SC-35), anti-GAPDH (1:5000, Sigma-Aldrich, St. Louis, MO, USA), and anti-β-tubulin. Subsequently, the blots were incubated with horseradish peroxidase (HRP-)-conjugated secondary antibody at room temperature for 1.5 h. The immunoreactive bands were visualized using enhanced chemiluminescence (ECL) (Thermo Fisher Scientific, Rockford, IL, USA). GAPDH or β-tubulin was used as the loading control.

### 4.14. Quantitative Real-Time PCR (qRT-PCR)

Total RNA was extracted from myocardial tissue using Trizol reagent (Takara, Kyoto, Japan), and then the purified RNA was reversely transcribed into cDNA using PrimeScript™ RT Master Mix Kit (Takara, Kyoto, Japan). SYBR Green Fast qPCR mix (Takara, Kyoto, Japan) and ABI 7500 real-time PCR system (ABI, Carlsbad, CA, USA) were used to perform qRT-PCR. All primers used were as follows: atrial natriuretic peptide (ANP)-F 5′-GAGAAGATGCCGGTAGAAGA-3′, ANP-R 5′-GAGAAGATGCCGGTAGAAGA-3′, brain natriuretic peptide (BNP)-F, 5′-CTGCTGGAGCTGATAAGAGA-3′, BNP-R 5′-TGCC CAAAGCAGCTTGAGAT-3′; CaMKIIδ A-F 5′-CGAGAAATTTTTCAGCAGCC-3′, CaMKIIδ A-R 5′-ACAGTAGTTTGGGGCTCCAG-3′; CaMKIIδ B-F 5′-CGAGAAATTTTTCAGCAGCC-3′, CaMKIIδ B-R 5′-GCTCTCAGTTGACTCCATCATC-3′; CaMKIIδ C-F 5′-CGAGAAATTTTTCAGCAGCC-3, CaMKIIδ C-R 5′-CTCAGTTGACTCCTTTACCCC-3′; 18S-F 5′-AGTCCCTGCCCTTTGTACACA-3′, 18S-R 5′-CGATCCGAGGGCCTCACTA-3′. 18S was used as a housekeeping gene. Each experiment was performed in triplicate. The relative expressions of target genes were calculated with the 2^−∆∆Ct^ method.

### 4.15. Myocardial Ultrastructure Examination

The fresh myocardia were cut into three pieces (1 mm) and fixed with 4% glutaraldehyde and 1% osmium acid. The samples were dehydrated with acetone, embedded in Epon812, stained with toluidine blue, cut into 70 nm sections, and stained with uranyl acetate and lead citrate. A transmission electron microscope (JEM-1230) was used to examine the ultrastructure of myocardial tissue. A visible image consisting of 15 randomly selected areas per slice was taken to determine the structure and number of mitochondria. The volume of mitochondria was calculated, and the number of mitochondria was counted.

### 4.16. Statistical Analysis

All data were expressed as mean ± standard error. GraphPad Prism 5.0 software was used to process the experimental data. Statistical analysis was performed using the unpaired Student’s *t*-test on comparisons between two groups and using the one-way ANOVA test followed by the Student-Newman-Keuls (SNK) test on comparisons among multiple groups. *p* < 0.05 was considered statistically significant.

## 5. Conclusions

In conclusion, our current findings proved that necroptosis played an important role in myocardial necroptosis in mice with myocardial hypertrophy. RIPK3 regulates necroptosis in hypertrophic cardiomyocytes by mediating two signaling pathways, RIPK3/RIPK1/MLKL and RIPK3/CaMKII. Additionally, we speculate that RIPK3 affects the expression of splicing factors ASF/SF2 and SC-35 to cause variable splicing disorder of CaMKIIδ in cardiomyocytes. RIPK3 deficiency could ameliorate myocardial injury in mice with cardiac hypertrophy, improve cardiac function, inhibit CaMKII activation, correct CaMKIIδ alternative splicing disorder, and reduce necrosis. Collectively, CaMKII activation and necroptosis were increased in myocardial hypertrophy in an RIPK3-dependent manner. These findings provided valuable insights into the pathology of myocardial hypertrophy.

## Figures and Tables

**Figure 1 ijms-24-14529-f001:**
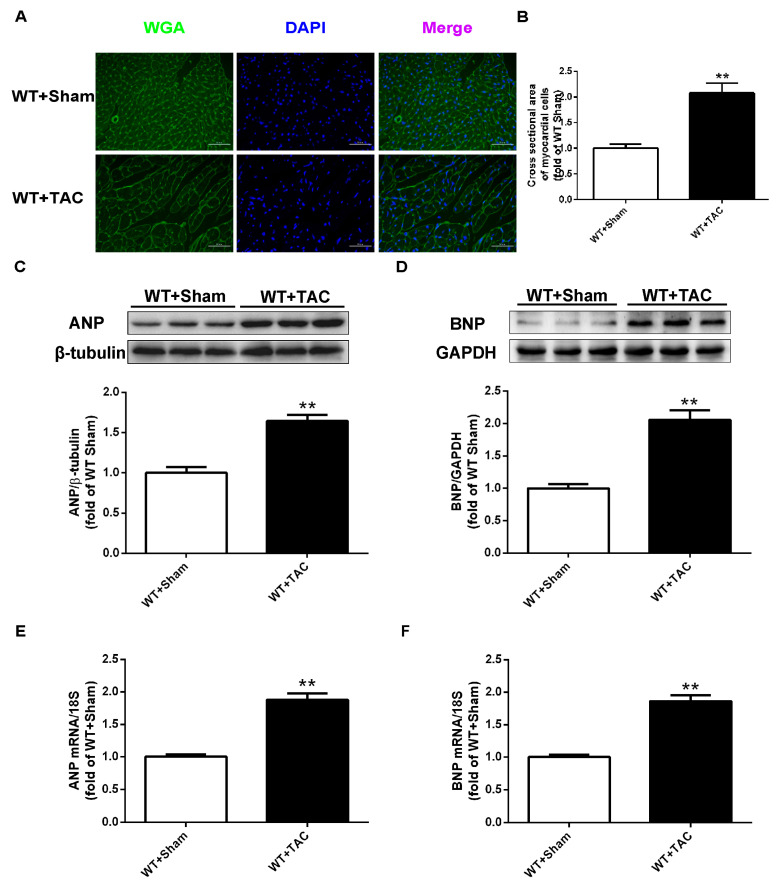
TAC induces an increased cross-sectional area of mouse cardiomyocytes and up-regulation of hypertrophy genes ANP and BNP. The model of myocardial hypertrophy was constructed by TAC surgery in male C57BL/6 mice, and a sham operation was performed in the control group. After 3 weeks, (**A**) WGA staining was used to detect cardiomyocyte hypertrophy. Bar = 50 μm. (**B**) Cell area measurements in WGA. (**C**,**D**) ANP and BNP were quantified by Western blotting analysis, with β-tubulin or GAPDH as a loading control. (**E**,**F**) The expressions of ANP and BNP at the mRNA level in the myocardium were detected by qRT-PCR. 18S serviced as a housekeeping gene. ** *p* < 0.01, significantly from WT + sham, n = 6.

**Figure 2 ijms-24-14529-f002:**
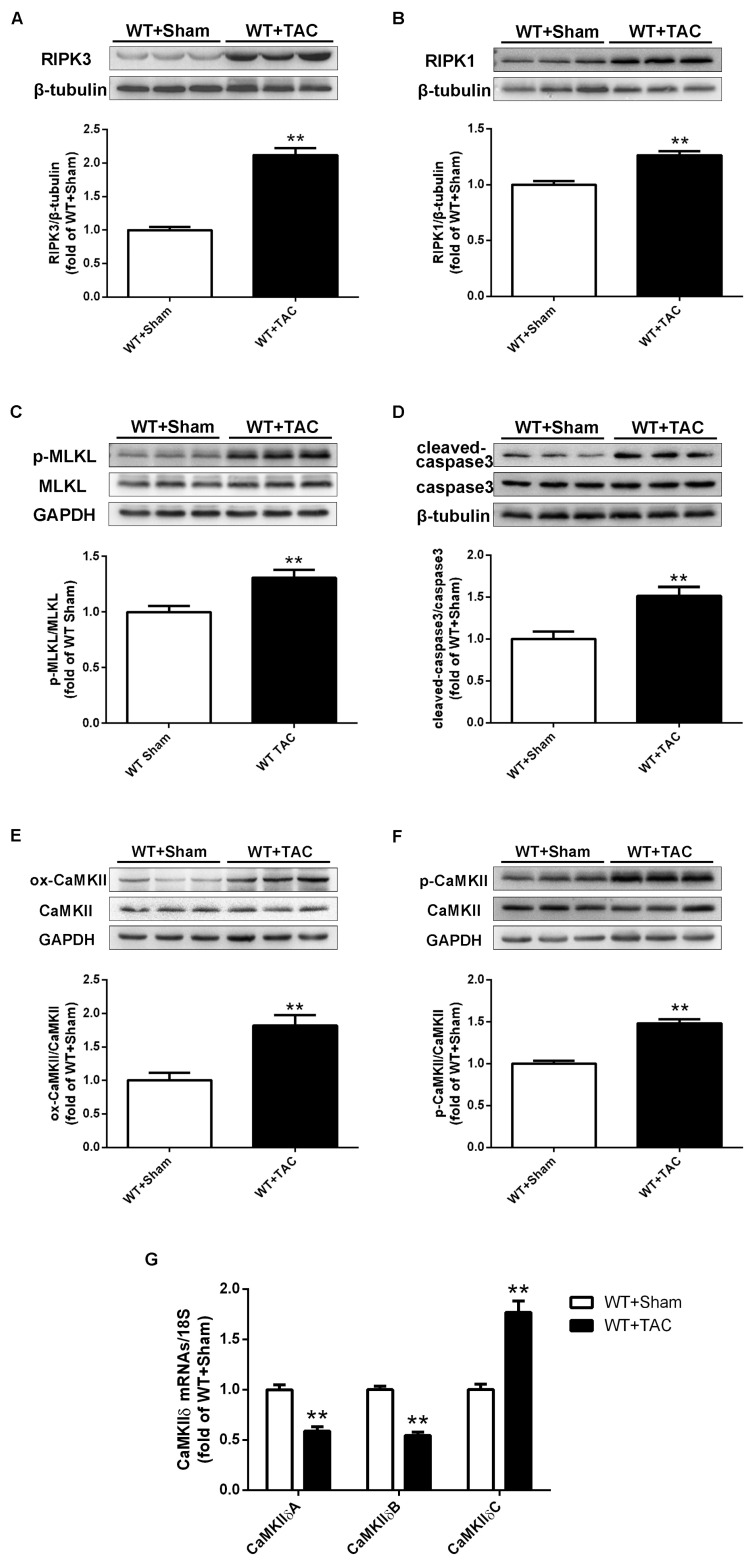
Mice with myocardial hypertrophy have necroptotic apoptosis and enhanced CaMKIIδ activity. At 3 weeks after TAC surgery, (**A**–**C**) the expressions of RIPK3, RIPK1, total MLKL, and MLKL phosphorylation were quantified by Western blotting analysis using β-tubulin or GAPDH as a loading control. (**D**) Apoptosis is a form of cell death. The total amounts of caspase-3 and cleaved caspase-3 were quantified by Western blotting analysis. (**E**,**F**) CaMKII oxidation (ox-CaMKII), CaMKII phosphorylation (p-CaMKII), and total CaMKII were assessed by Western blotting analysis. (**G**) The expressions of CaMKIIδ A, CaMKIIδ B, and CaMKIIδ C at the mRNA level in the myocardium were detected by qRT-PCR. 18S serviced as a housekeeping gene. ** *p* < 0.01, significantly from WT + sham, n = 6.

**Figure 3 ijms-24-14529-f003:**
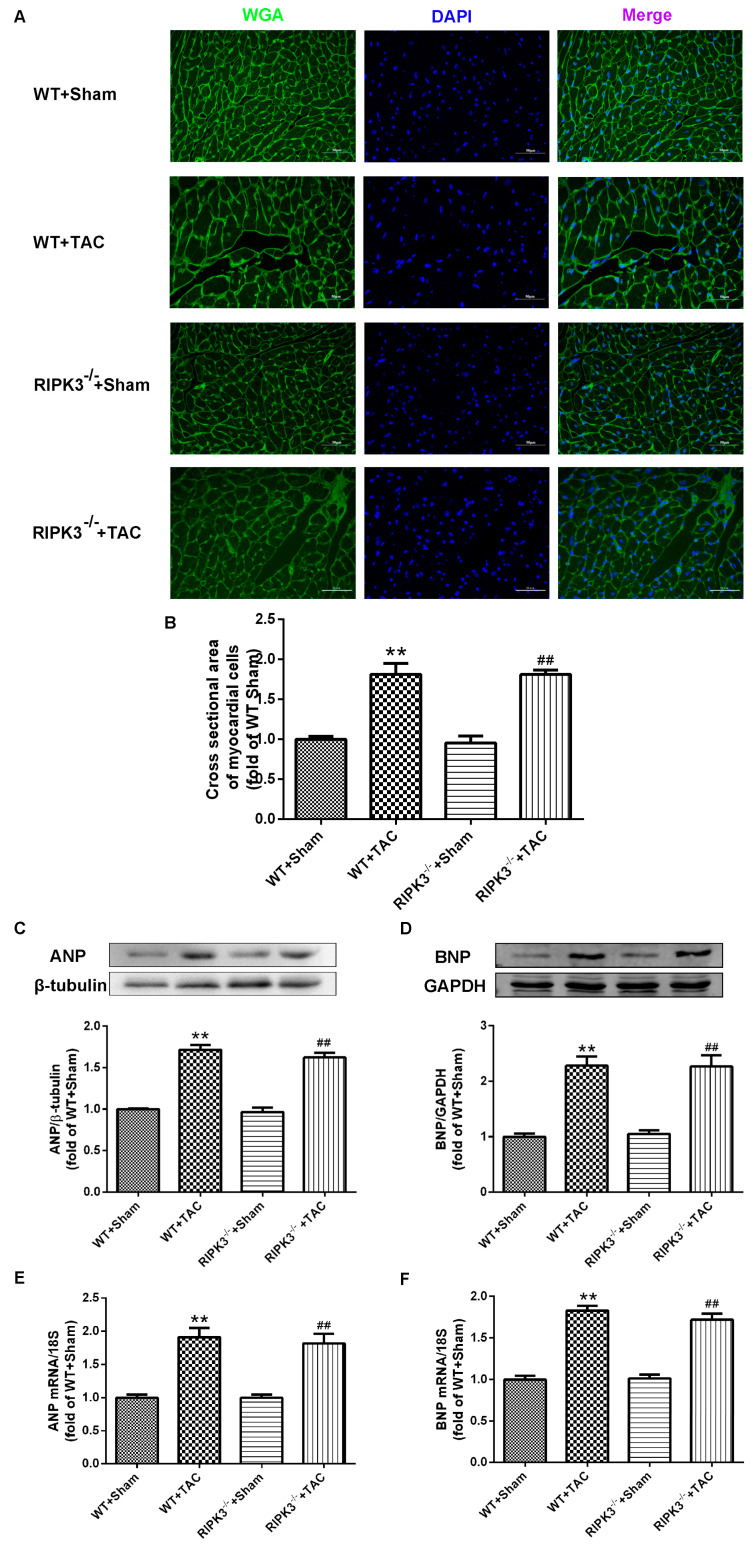
TAC is used in WT and RIPK3^−/−^ mice to construct a myocardial hypertrophy model, and the control group undergoes a sham operation. After 3 weeks, (**A**) cardiomyocyte hypertrophy was detected by WGA staining. Bar = 50 μm. (**B**) Cell area measurements in WGA. (**C**,**D**) ANP and BNP were quantified by Western blotting analysis. (**E**,**F**) The expressions of ANP and BNP at the mRNA level in the myocardium were detected by qRT-PCR. 18S serviced as a housekeeping gene. ** *p* < 0.01, significantly from WT + sham; ^##^ *p* < 0.01 significantly from RIPK3^−/−^ + sham, n = 6.

**Figure 4 ijms-24-14529-f004:**
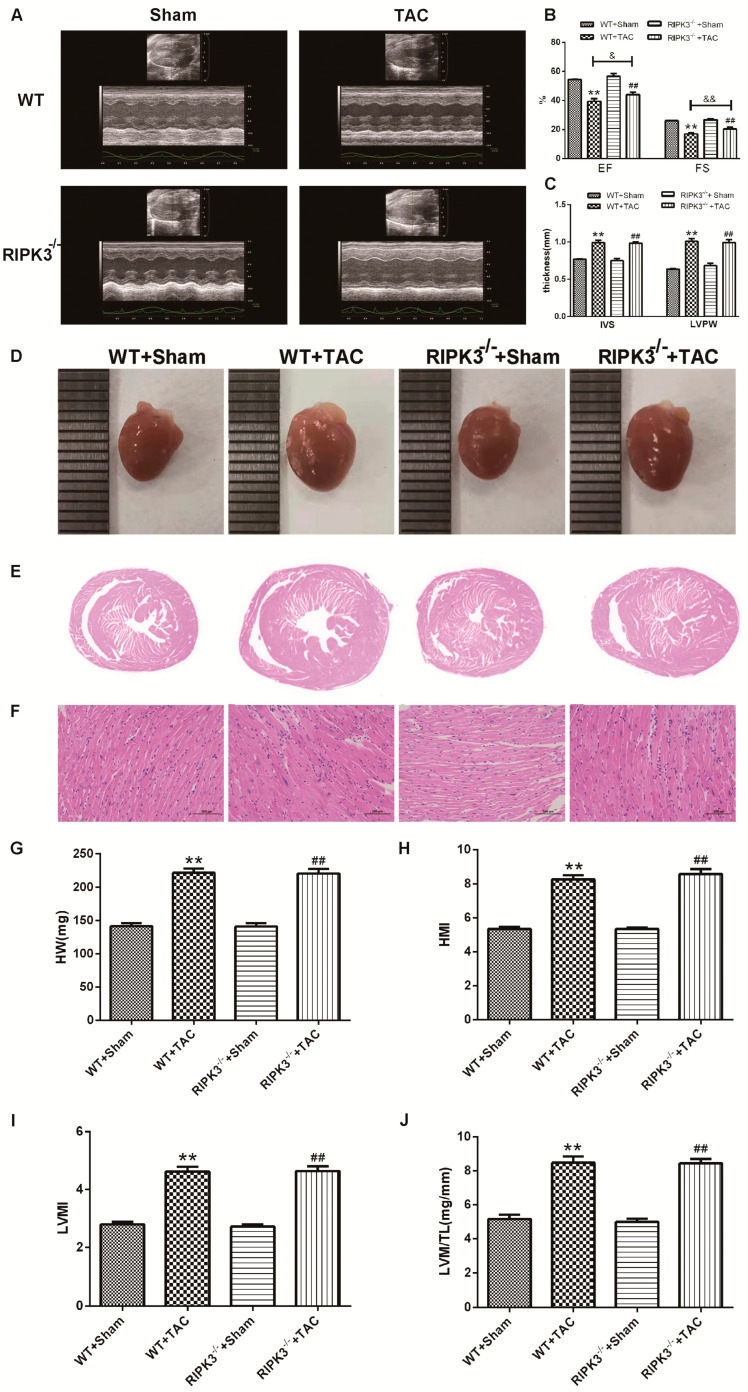
RIPK3 deficiency can alleviate cardiac dysfunction in mice with myocardial hypertrophy and improve the disorder of myocardial cell arrangement. At 3 weeks after TAC surgery, (**A**–**C**) cardiac function was assessed by echocardiography, and EF, FS, IVS, and LVPW were calculated. (**D**) The epigraph of the heart. (**E**) Myocardial tissue examination under a microscope. (**F**) Myocardium injury was measured by H&E staining. Bar = 50 μm. (**G**–**J**) The heart indexes HW, HMI, LVMI, and LVW/TL ratio (LVW/TL) were calculated by measurement analysis. ** *p* < 0.01 significantly from WT + sham; ^##^ *p* < 0.01 significantly from RIPK3^−/−^ + sham and ^&^ *p* < 0.05, ^&&^ *p* < 0.01 significantly from WT + TAC, n = 6.

**Figure 5 ijms-24-14529-f005:**
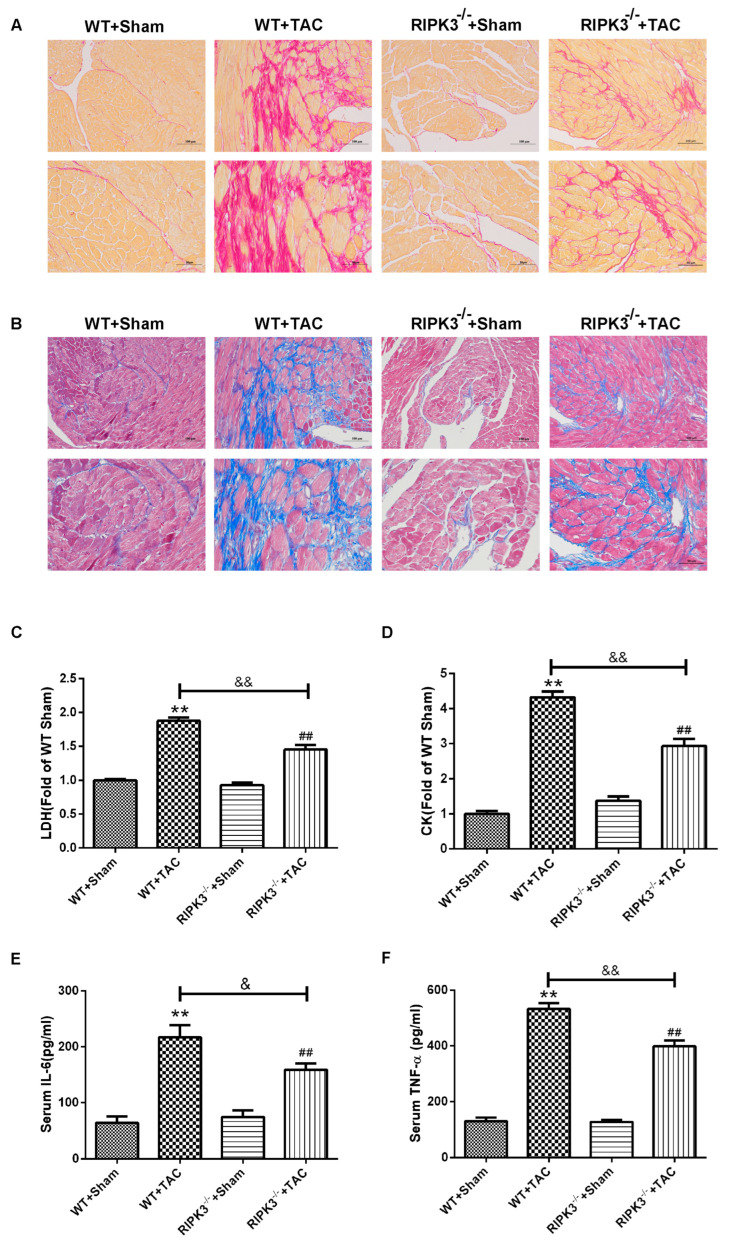
RIPK3 deficiency can reduce myocardial injury, myocardial fibrosis, and inflammation in mice with myocardial hypertrophy. At 3 weeks after TAC surgery, (**A**) myocardial collagen deposition was detected by the Sirius scarlet staining method. Scale bar: The superscript is 100 μm, and the subscript is 50 μm. (**B**) Myocardial collagen deposition was detected by the Masson staining method. Scale bar: The superscript is 100 μm, and the subscript is 50 μm. (**C**,**D**) The levels of serum LDH and CK were used to evaluate the degree of myocardial injury. (**E**,**F**) Serum IL-6 and TNF-α levels were detected by ELISA ** *p* < 0.01, significantly from WT + sham; ^##^ *p* < 0.01 significantly from RIPK3^−/−^ + sham and ^&^ *p* < 0.05, ^&&^ < 0.01 significantly from WT + TAC, n = 6.

**Figure 6 ijms-24-14529-f006:**
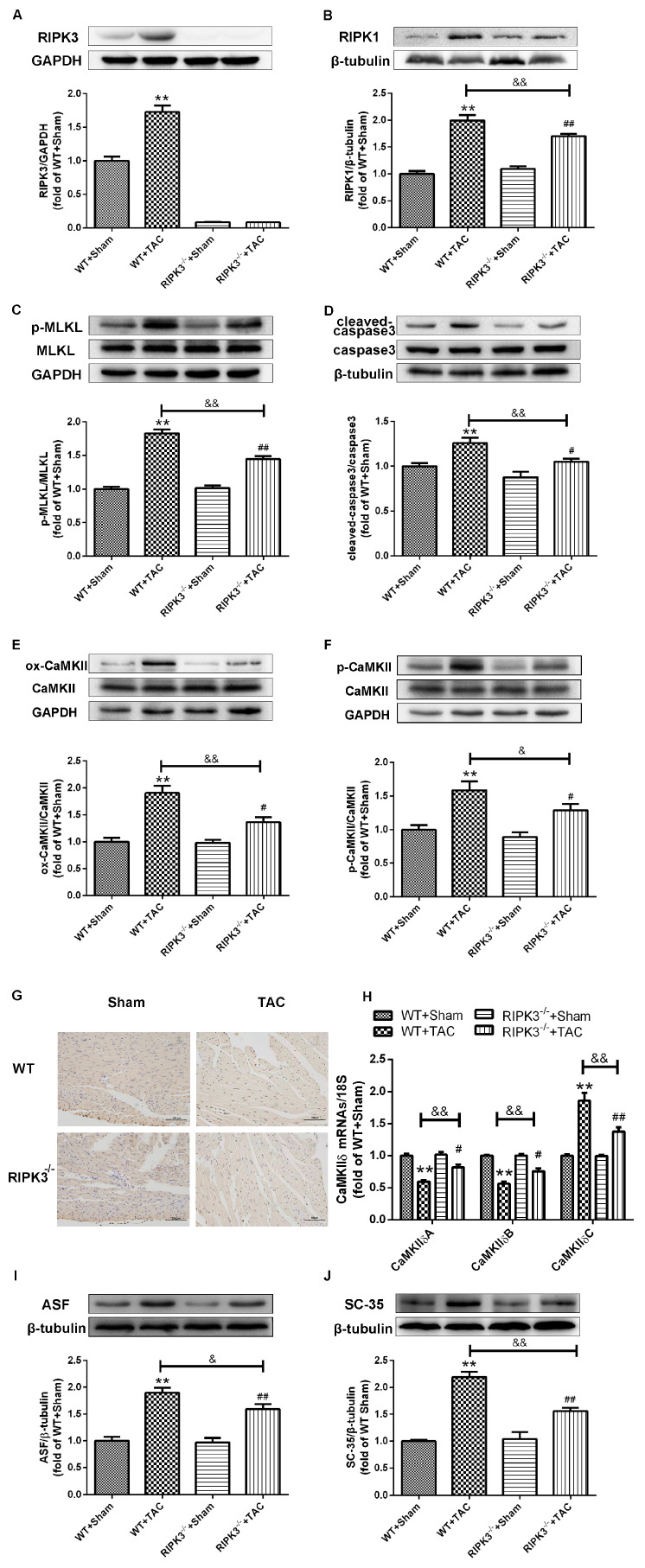
Loss of RIPK3 can ameliorate myocardial necroptosis and correct CaMKIIδ splicing disorder in mice with myocardial hypertrophy. (**A**) Detection by Western blotting analysis showed that the myocardial tissue of RIPK3^−/−^ mouse did not express RIPK3. β-tubulin was used as a loading control. (**B**–**D**) Western blotting analysis was used to detect the expression of RIPK1, the total amount of MLKL and MLKL phosphorylation, the total amount of caspase-3, and the expression of cleaved caspase-3 in myocardial tissue. β-tubulin was used as a loading control. (**E**,**F**) CaMKII oxidation (ox-CaMKII), CaMKII phosphorylation (p-CaMKII), and the total amount of CaMKII were determined by Western blotting analysis. β-tubulin or GAPDH was used as a loading control. (**G**) Cell apoptosis of the myocardium was detected with TUNEL staining. (**H**) The expressions of CaMKIIδ A, CaMKIIδ B, and CaMKIIδ C at the mRNA level in the myocardium were detected by qRT-PCR. 18S serviced as a housekeeping gene. (**I**,**J**) The expressions of ASF and SC-35 were detected by Western blotting analysis. β-tubulin was used as a loading control. ** *p* < 0.01, significantly from WT + sham; ^#^
*p* < 0.05, ^##^
*p* < 0.01 significantly from RIPK3^−/−^ + sham and ^&^
*p* < 0.05, ^&&^
*p* < 0.01 significantly from WT + TAC, n = 6.

**Figure 7 ijms-24-14529-f007:**
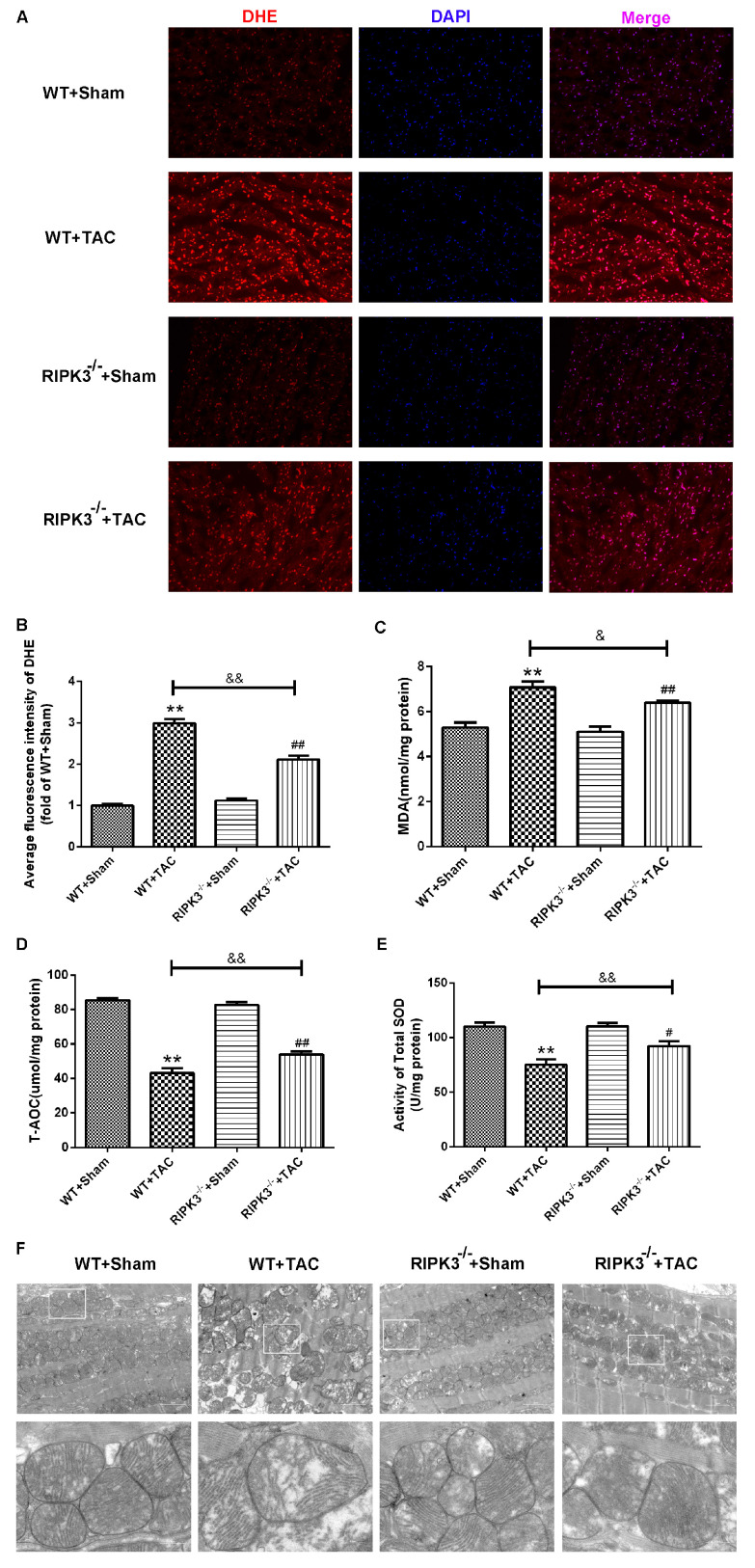
Loss of RIPK3 can improve myocardial oxidative stress and mitochondrial ultrastructure in mice with myocardial hypertrophy. At 3 weeks after the TAC surgery, (**A**) DHE fluorescent probe was used to detect myocardial superoxide production under a fluorescent microscope. (**B**) Average fluorescence intensity measurements of DHE. (**C**–**E**) Determination of myocardial MDA, T-AOC, and T-SOD activity. (**F**) Myocardial mitochondrial ultrastructure was examined with a transmission electron microscope. Scale bar: superscript is 2 μm, and a subscript is 0.5 μm. ** *p* < 0.01, significantly from WT + sham; ^#^ *p* < 0.05, ^##^ *p* < 0.01 significantly from RIPK3^−/−^ + sham and ^&^ *p* < 0.05, ^&&^ *p* < 0.01 significantly from WT + TAC, n = 6.

**Figure 8 ijms-24-14529-f008:**
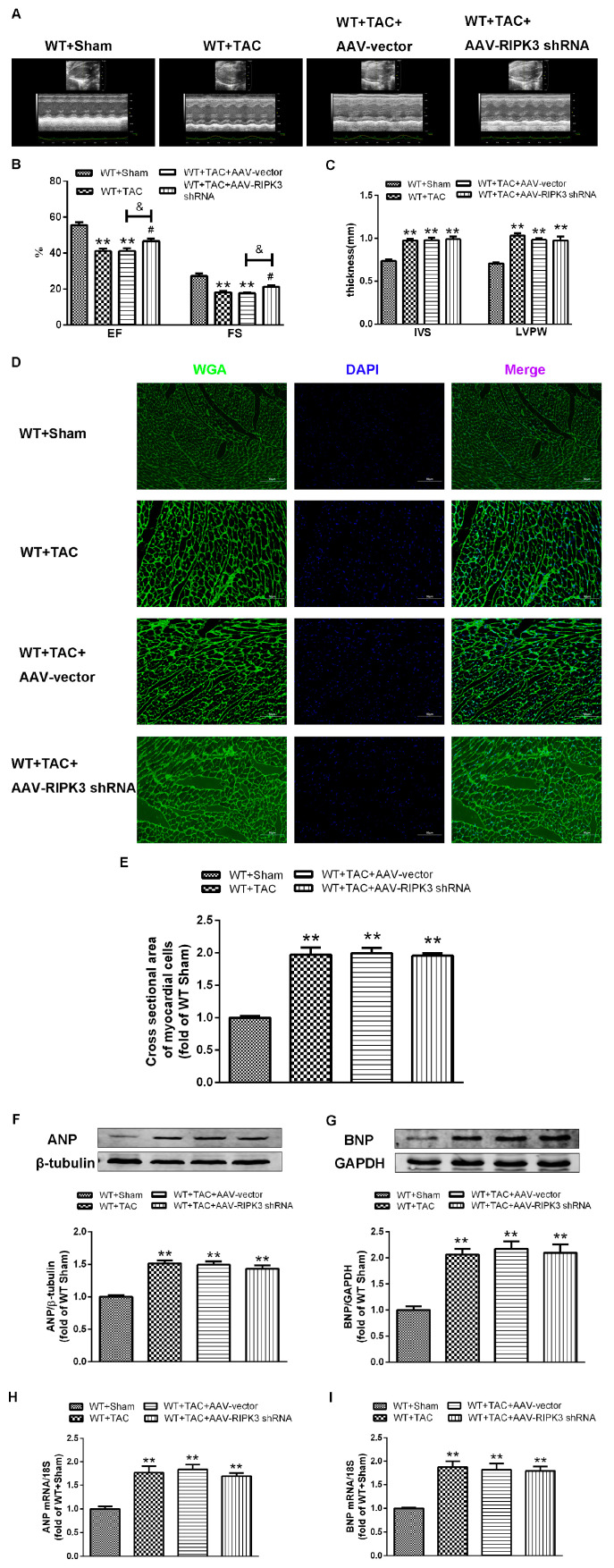
A myocardial hypertrophy model is established by TAC after AAV-shRNA pretreatment. After 3 weeks (**A**–**C**), we used echocardiography to assess cardiac function and calculate EF, FS, IVS, and LVPW values. (**D**) Cardiomyocyte hypertrophy was detected by WGA staining. Bar = 50 μm. (**E**) Cell area measurements in WGA. (**F**,**G**) ANP and BNP were quantified by Western blotting analysis. (**H**,**I**) The expressions of ANP and BNP at the mRNA level in the myocardium were detected by qRT-PCR. 18S serviced as a housekeeping gene. ** *p* < 0.01, significantly from WT + sham; ^#^ *p* < 0.05 significantly from WT + TAC and ^&^ *p* < 0.05 significantly from WT + TAC + AAV-vector, n = 6.

**Figure 9 ijms-24-14529-f009:**
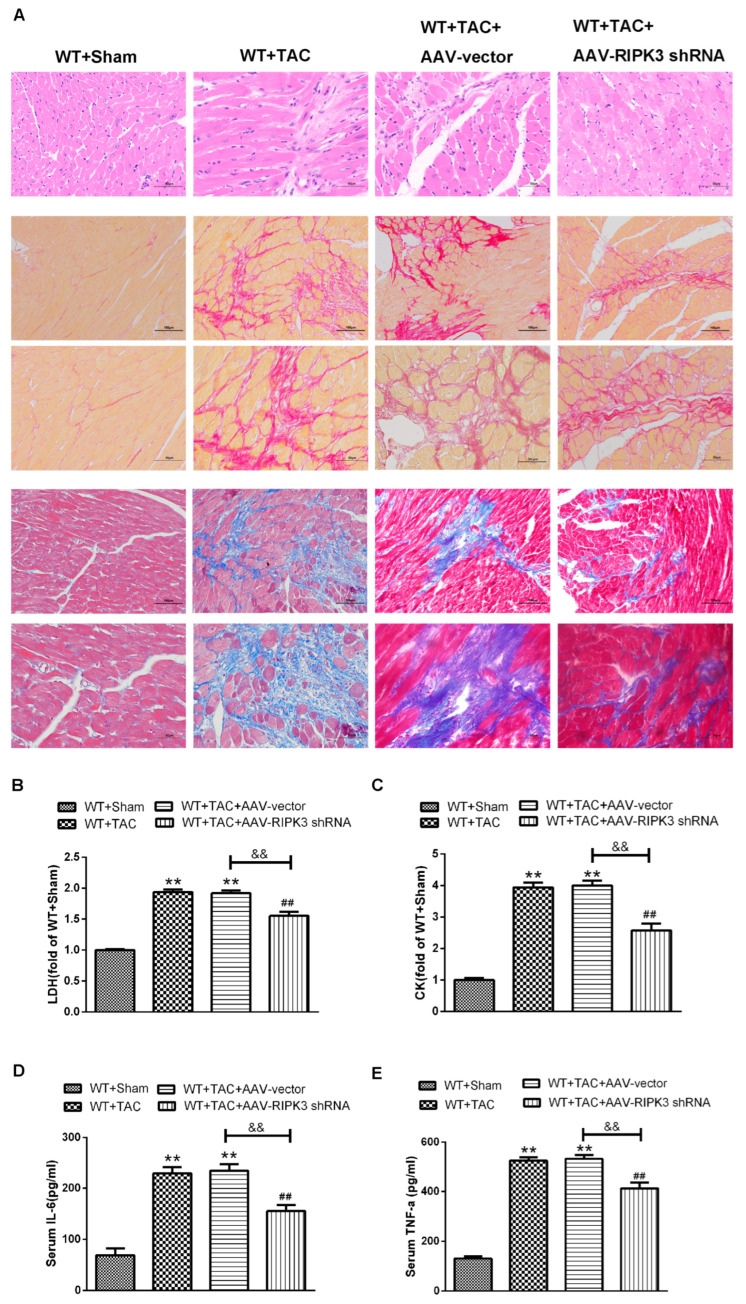
AAV-shRNA pretreatment improves myocardial injury, myocardial fibrosis, and inflammation in mice with myocardial hypertrophy. (**A**) Myocardium injury was measured by H&E staining. Bar = 50 μm. Sirius’ scarlet staining method and the Masson staining method were used to determine myocardial collagen deposition. Scale bar: The superscript is 100 μm, and the subscript is 50 μm. (**B**,**C**) The kits were used to detect serum LDH and CK levels to evaluate the degree of myocardial injury. (**D**,**E**) Serum IL-6 and TNF-α levels were detected by ELISA. ** *p* < 0.01, significantly from WT + sham; ^##^
*p* < 0.01 significantly from WT + TAC and ^&&^
*p* < 0.01 significantly from WT + TAC + AAV-vector, n = 6.

**Figure 10 ijms-24-14529-f010:**
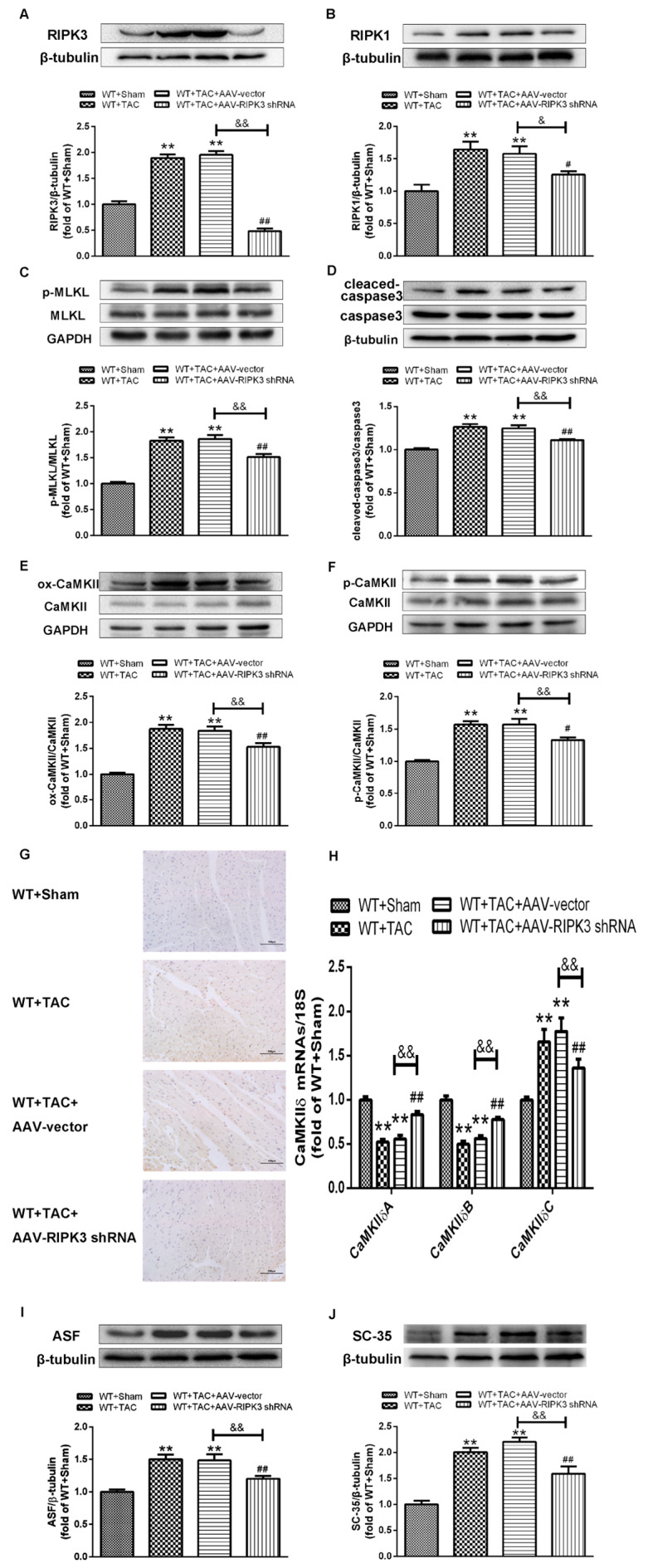
After pretreatment with AAV-shRNA, the expression of RIPK3 is down-regulated, myocardial necroptosis is improved in mice with myocardial hypertrophy, and CaMKIIδ splicing disorder is corrected. (**A**) Western blotting analysis showed that the expression of RIPK3 in mouse myocardium was decreased after AAV-shRNA pretreatment. β-tubulin was used as a loading control. (**B**–**F**) The expressions of RIPK1, total MLKL, and MLKL phosphorylation, total caspase-3 and cleaved caspase-3, CaMKII oxidation (ox-CaMKII), CaMKII phosphorylation (p-CaMKII), and total CaMKII were detected by Western blotting analysis. (**G**) Cell apoptosis of myocardium was detected with TUNEL staining. (**H**) The expressions of CaMKIIδ A, CaMKIIδ B, and CaMKIIδ C at the mRNA level in the myocardium were detected by qRT-PCR. 18S serviced as a housekeeping gene. (**I**,**J**) The expressions of ASF and SC-35 were detected by Western blotting analysis. β-tubulin was used as a loading control. ** *p* < 0.01, significantly from WT + sham; ^#^
*p* < 0.05, ^##^
*p* < 0.01 significantly from WT + TAC and ^&^
*p* < 0.05, ^&&^
*p* < 0.01 significantly from WT + TAC + AAV-vector.

**Figure 11 ijms-24-14529-f011:**
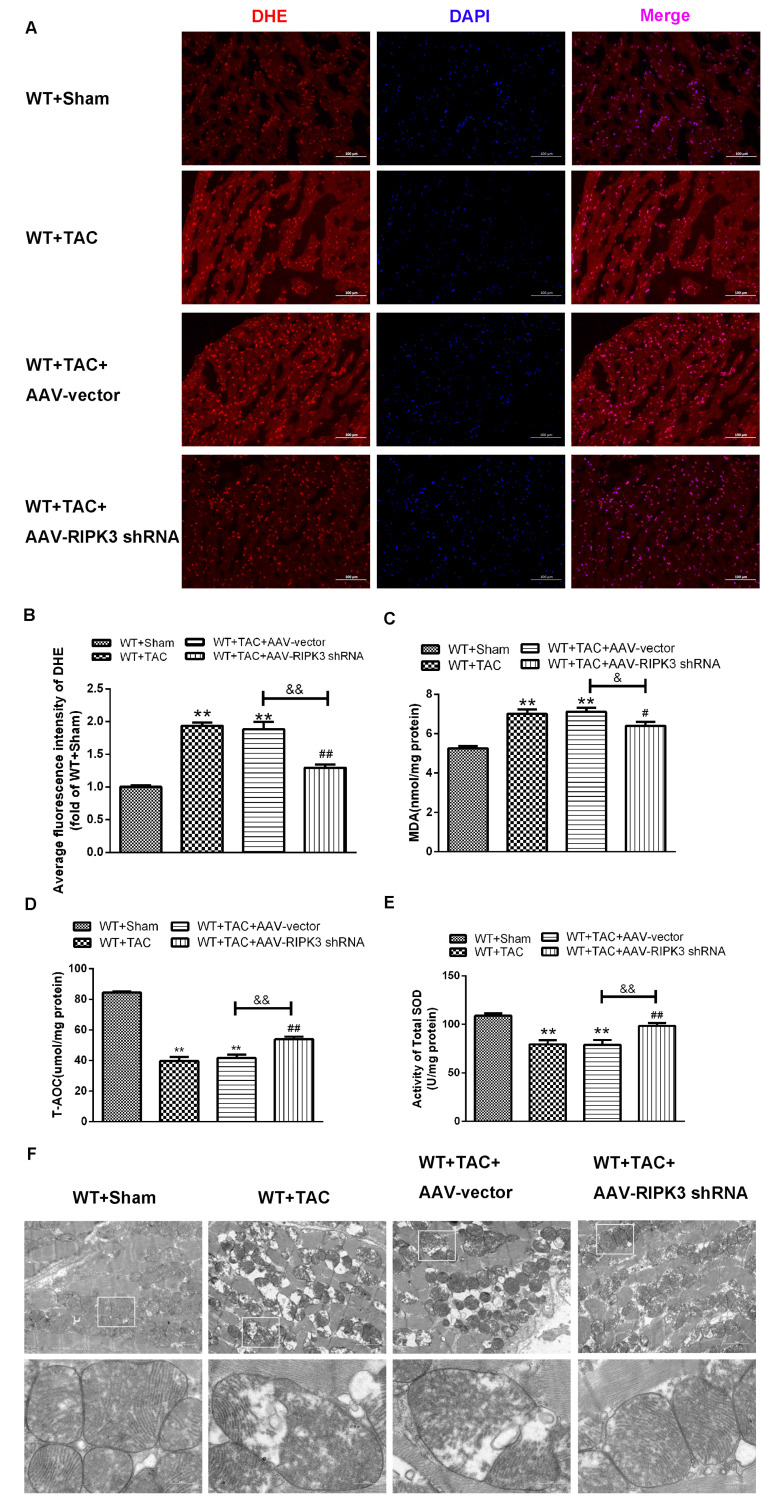
AAV-shRNA pretreatment improves myocardial oxidative stress and mitochondrial ultrastructure. At 3 weeks after the TAC surgery, (**A**) DHE fluorescent probe was used to detect myocardial superoxide production under a fluorescent microscope. (**B**) Average fluorescence intensity measurements of DHE. (**C**–**E**) Determination of myocardial MDA, T-AOC, and T-SOD activity. (**F**) Myocardial mitochondrial ultrastructure was examined with a transmission electron microscope. Scale bar: The superscript is 2 μm, and the subscript is 0.5 μm. ** *p* < 0.01, significantly from WT + sham; ^#^
*p* < 0.05, ^##^
*p* < 0.01 significantly from WT + TAC and ^&^
*p* < 0.05, ^&&^
*p* < 0.01 significantly from WT + TAC + AAV-vector, n = 6.

## Data Availability

The data and material used to support the findings of this study are available from the corresponding authors upon request.

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
