# Peer review of "The Regulatory Effect of Receptor-Interacting Protein Kinase 3 on CaMKIIδ in TAC-Induced Myocardial Hypertrophy"

_ijms, 2023, doi:10.3390/ijms241914529_

Round 1

Reviewer 1 Report

Myocardial hypertrophy is a prevalent disease that is induced by pressure and volume overload. However, up to now, the molecular mechanism that are involved in it are not fully understood. Previous studies have demonstrated that in diseased myocardial tissue, may undergo necroptosis that is induced by RIPK1/RIPK3/MLKL cascade which is regulated by the RIPK3/CaMKII signaling. The current study aimed to explore the role and mechanism of RIPK3-mediated necroptosis in pressure overload-induced cardiac hypertrophy. The authors used transverse aortic constriction on wild-type and knockout mice to establish a model of myocardial hypertrophy, and after 3 weeks measured various parameters involved in the process.  Their results indicate that RIPK3 deficiency can alleviate cardiac hypertrophic effects, acting via CaMKII activation. These results proved that RIPK3 is a target for therapeutic intervention of myocardial hypertrophy. Overall, this is a well-written, comprehensive, and important study that establish the role of RIPK3 and CaMKII in myocardial hypertrophy and validate RIPK3 as a therapeutic target for this disease. I have only a few minor comments as follows: 

1)     The results derived by the WGA staining in figures 1, 3, 8 are not very convincing. Is there a way to quantify them? 

2)     The same is true for the DHE staining in Figs. 7 and 11. 

3)     The DAPI staining in Fig. 7 and 11 is not very clear, and the authors do not relate to it. This should be corrected. 

4)     The manuscript is generally well written, but the abstract is somewhat confusing. It is recommended to make it better understood. 

5) The authors are using the term necrosis and necroptosis interchangeably. Is there any reason for it? It is also not clear why is the Material and Method section divided into two parts (Section 2 and Section 3).

6) In the discussion, it is recommended to include a scheme with the suggested signaling pathway leading to necroptosis in Myocardial hypertrophy. 

Generally fine.

Author Response

Thank you for sending us the insightful and specific comments on our manuscript, which are valuable for revising and improving our manuscript. We have reviewed the comments carefully and made corresponding revisions. We hope the revised manuscript will now meet the requirements of the journal. Revisions in the manuscript are marked in red, and our detailed point-by-point responses to your comments are listed below.

Reviewer#1:

1) The results derived by the WGA staining in figures 1, 3, 8 are not very convincing. Is there a way to quantify them? 

Response:Thank you for your suggestion.We have added a new bar chart to the figures 1, 3, 8 to analyze and compare the cross-sectional area of myocardial cells. Revisions in the manuscript are marked in red.

2) The same is true for the DHE staining in Figs. 7 and 11.

Response:Thank you for your suggestion.We analyzed and compared the fluorescence intensity of DHE staining. Then we added the results to the Figures7B, 11B. Revisions in the manuscript are marked in red.

3)  The DAPI staining in Fig. 7 and 11 is not very clear, and the authors do not relate to it. This should be corrected.

Response:Thank you for your suggestion. We have made revisions to the relevant images.Revisions in the manuscript are marked in red.

4)  The manuscript is generally well written, but the abstract is somewhat confusing. It is recommended to make it better understood. 

Response:Thank you for your advice. We have supplemented and modified the abstract section and explained CaMKII δ Possible mechanisms of variable splicing disorder.Revisions in the manuscript are marked in red.

5) The authors are using the term necrosis and necroptosis interchangeably. Is there any reason for it? It is also not clear why is the Material and Method section divided into two parts (Section 2 and Section 3).

Response: Thank you very much for pointing it out. This article mainly discusses the role of myocardial cell necroptosis in myocardial hypertrophy, therefore we have revised the term 'necrosis' in the article to' necroptosis '. Revisions in the manuscript are marked in red. In addition, the Section 2 of this article is about Materials and Methods, and the Section 3 is about Results.

6) In the discussion, it is recommended to include a scheme with the suggested signaling pathway leading to necroptosis in Myocardial hypertrophy.

Response: Thank you for your advice. In the discussion section, we have supplemented the signaling pathways that lead to necrotic apoptosis of hypertrophic cardiomyocytes.Revisions in the manuscript are marked in red.

kind regards.
Your sincerely

Wei Zhang

Reviewer 2 Report

Summary

In the present study, Qian et al. have invested the role and mechanism of RIPK3-mediated necroptosis in pressure overload-induced cardiac hypertrophy using transverse aortic constriction (TAC) surgery model in mice to explore the effects of RIPK3 deficiency and downregulation on myocardial hypertrophy. They also showed that silencing and downregulating  RIPK3 could regulate CaMKIIδ alternative splicing and CaMKII activity to delay the pathogenesis of cardiac hypertrophy. The study and its findings have high significance in the field.

While carefully reviewing the manuscript, some concerns require attention and comments that need to be addressed.

1.       A previous study reported, “During excitation-contraction coupling (ECC) CaMKII phosphorylates several Ca-handling proteins including ryanodine receptors (RyR), phospholamban, and L-type Ca channels. CaMKII expression and activity have been shown to correlate positively with impaired ejection fraction in the myocardium of patients with heart failure, and CaMKII has been proposed to be a possible compensatory mechanism to keep hearts from complete failure. However, in addition to these acute effects on ECC, CaMKII was shown to be involved in hypertrophic signaling, termed excitation-transcription coupling (ETC). The authors should discuss these details in their manuscript (PMID: 16138211).

2.       Previous studies on animal models have shown that overexpression of nuclear isoform CaMKIIdelta B can induce myocyte hypertrophy. However, the authors didn’t discuss this information in the current manuscript.

3.       A previous Hong Kong Xue 2022 study has already demonstrated “RIP3 Contributes to Cardiac Hypertrophy by Influencing MLKL-Mediated Calcium Influx.” The authors should discuss their findings in the context of Xue et al.

4.       A very similar study by Cao et al. has shown that “The regulatory effect of RIPK3 on CaMKII δ alternative splicing and CaMKII activity to ameliorate transverse arch constriction (TAC)-induced myocardial necroptosis in wild type (WT) and RIPK3–/– mice with HF (Cao et al., 2022). How the current study is novel? Moreover, this study was not cited in the reference. Why?

Cao J, Zhang J, Qian J, Wang X, Zhang W, Chen X. Ca2+/Calmodulin-Dependent Protein Kinase II Regulation by RIPK3 Alleviates Necroptosis in Transverse Arch Constriction-Induced Heart Failure. Front Cardiovasc Med. 2022 Apr 28;9:847362. doi: 10.3389/fcvm.2022.847362. PMID: 35571197; PMCID: PMC9097920.

5.       Regulation of Camklld by RIPK3 is also demonstrated earlier by Hua et al. 2022. The Authors should discuss this in their manuscript. What are the novel insights in their study as compared to this previous study?

Hua Y, Qian J, Cao J, Wang X, Zhang W, Zhang J. Ca2+/Calmodulin-Dependent Protein Kinase II Regulation by Inhibitor of Receptor Interacting Protein Kinase 3 Alleviates Necroptosis in Glycation End Products-Induced Cardiomyocytes Injury. Int J Mol Sci. 2022 Jun 23;23(13):6988. doi: 10.3390/ijms23136988. PMID: 35805993; PMCID: PMC9266390.

Minor comments

1.       Line 124, which suture was used for closing the thoraces.

2.       In the tail vein injection, how long did the authors wait after AAV injection before performing the surgery?

3.       For ultrasound recording, which probe was utilized? Describe the instrument's details.

4.       Has the author used 1-2% isoflurane for induction and maintenance of anesthesia?

5.       How long it took for the fibrosis to occur after TAC surgery?

6.       Figures 1A, 3A, 7A, 6G, 10G, 11A, should be supplemented by quantification wherever possible to support the comparisons.

7.       Figure 6C requires quantification to support the fold change after TAC surgery.

Author Response

Thank you for sending us the insightful and specific comments on our manuscript, which are valuable for revising and improving our manuscript. We have reviewed the comments carefully and made corresponding revisions. We hope the revised manuscript will now meet the requirements of the journal. Revisions in the manuscript are marked in red, and our detailed point-by-point responses to your comments are listed below.

Reviewer#2:

  1. A previous study reported, “During excitation-contraction coupling (ECC) CaMKII phosphorylates several Ca-handling proteins including ryanodine receptors (RyR), phospholamban, and L-type Ca channels. CaMKII expression and activity have been shown to correlate positively with impaired ejection fraction in the myocardium of patients with heart failure, and CaMKII has been proposed to be a possible compensatory mechanism to keep hearts from complete failure. However, in addition to these acute effects on ECC, CaMKII was shown to be involved in hypertrophic signaling, termed excitation-transcription coupling (ETC). The authors should discuss these details in their manuscript (PMID: 16138211).

Response: Thank you for your suggestion.These details were added to the discussion. Please refer to the revised draft for details. Revisions in the manuscript are marked in red.

  1. Previous studies on animal models have shown that overexpression of nuclear isoform CaMKIIdelta B can induce myocyte hypertrophy. However, the authors didn’t discuss this information in the current manuscript.

Response:Thank you for your suggestion. Previous studies on animal models have shown that overexpression of nuclear isoform CaMKIIdelta B can induce myocyte hypertrophy.However,other studies showed that CaMKIIdelta Ccan induce myocyte hypertrophy too.Our experimental results found that the expression of CaMKIIdelta C increased in myocardial hypertrophy, while the expression of CaMKIIdelta B decreased in myocardial hypertrophy.Therefore, this may be an issue that needs to be further explored.

  1. A previous Hong Kong Xue 2022 study has already demonstrated “RIP3 Contributes to Cardiac Hypertrophy by Influencing MLKL-Mediated Calcium Influx.” The authors should discuss their findings in the context of Xue et al

Response: Thank you for your suggestion.These details were added to the discussion. At the same time, references were added and the reference numbers were adjusted.Please refer to the revised draft for details. Revisions in the manuscript are marked in red.

  1. A very similar study by Cao et al. has shown that “The regulatory effect of RIPK3 on CaMKII δ alternative splicing and CaMKII activity to ameliorate transverse arch constriction (TAC)-induced myocardial necroptosis in wild type (WT) and RIPK3–/– mice with HF (Cao et al., 2022). How the current study is novel? Moreover, this study was not cited in the reference. Why?

Response:Thank you for your suggestion.Cao et al.'s research involves the relevant mechanisms of myocardial cell necrotic apoptosis after myocardial hypertrophy develops into heart failure. Before we completed this manuscript, its article had not yet been published, so we did not cite this study. Our study further explores the possible mechanisms that may cause CaMKIIdelta variable splicing based on this.

  1. Regulation of Camklld by RIPK3 is also demonstrated earlier by Hua et al. 2022. The Authors should discuss this in their manuscript. What are the novel insights in their study as compared to this previous study?

Hua Y, Qian J, Cao J, Wang X, Zhang W, Zhang J. Ca2+/Calmodulin-Dependent Protein Kinase II Regulation by Inhibitor of Receptor Interacting Protein Kinase 3 Alleviates Necroptosis in Glycation End Products-Induced Cardiomyocytes Injury. Int J Mol Sci. 2022 Jun 23;23(13):6988. doi: 10.3390/ijms23136988. PMID: 35805993; PMCID: PMC9266390.

Response:Hua et alfound AGEs increased the expression of RIPK3, aggravated the disorder of CaMKII δ alternative splicing, promoted CaMKII activation, enhanced oxidative stress, induced necroptosis, and damaged cardiomyocytes.RIPK3 downregulation corrected CaMKII δ alternative splicing disorder, inhibited CaMKII activation, reduced oxidative stress, attenuated necroptosis, and improved cell damage in cardiomyocytes.Based on their research findings, we associate whether RIPK3 has a similar regulatory effect on CaMKII in myocardial hypertrophy diseases, and further explore the possible mechanism of RIPK3 causing CaMKIIdelta variable splicing disorder.

Minor comments

  1. Line 124, which suture was used for closing the thoraces.

Response: Thank you for your advice. At the end of the operation, the chest was closed with 5-0 nylon sutures. Revisions in the manuscript are marked in red.

  1. In the tail vein injection, how long did the authors wait after AAV injection before performing the surgery?

Response: Thank you for your advice.After one week of injection into the tail vein, the adeno-associated virus showed better infection of mouse myocardial cells. Therefore,we wait one week after AAV injection before performing the surgery.Revisions in the manuscript are marked in red.

  1. For ultrasound recording, which probe was utilized? Describe the instrument's details.

Response:We used the Visual Sonic Vevo 2100 ultrasound diagnostic device to detect cardiac configuration through the long axis view of the sternum, and recorded images using M-mode echocardiography.

  1. Has the author used 1-2% isoflurane for induction and maintenance of anesthesia?

Response:Yes, we used 1-2% isoflurane for induction and maintenance of anesthesia. Revisions in the manuscript are marked in red.

  1. How long it took for the fibrosis to occur after TAC surgery?

Response:Three weeks after surgery, fibrosis was found in the myocardial tissue of mice with hypertrophic myocardium through Sirius red staining and Masson staining.

  1. Figures 1A, 3A, 7A, 6G, 10G, 11A, should be supplemented by quantification wherever possible to support the comparisons.

Response: Thank you very much for pointing it out.We quantified and analyzed the WGA staining maps of Figures 1A ,3A, as well as the DHE staining maps of Figures 7A ,11A, and added the statistical maps to the original images.

  1. Figure 6C requires quantification to support the fold change after TAC surgery.

Response: Thank you for your advice. We have quantifiedFigure 6C.

kind regards.
Your sincerely

Wei Zhang

Round 2

Reviewer 2 Report

Thank You for all the responses and adequate information provided. The comments have been addressed and there are no further concerns. I found the manuscript to be improved after the revisions. Both the reviewer's comments were addressed appropriately.